# Cost-Effective Cultivation of Native PGPB *Sinorhizobium* Strains in a Homemade Bioreactor for Enhanced Plant Growth

**DOI:** 10.3390/bioengineering10080960

**Published:** 2023-08-13

**Authors:** Luis Alberto Manzano-Gómez, Reiner Rincón-Rosales, José David Flores-Felix, Adriana Gen-Jimenez, Víctor Manuel Ruíz-Valdiviezo, Lucia María Cristina Ventura-Canseco, Francisco Alexander Rincón-Molina, Juan José Villalobos-Maldonado, Clara Ivette Rincón-Molina

**Affiliations:** 1Laboratorio de Ecología Genómica, Tecnológico Nacional de México, Instituto Tecnológico de Tuxtla Gutiérrez, Tuxtla Gutiérrez 29050, Chiapas, Mexico; luismg@3rbiotec.com (L.A.M.-G.); reiner.rr@tuxtla.tecnm.mx (R.R.-R.); d10270415@tuxtla.tecnm.mx (A.G.-J.); victor.rv@tuxtla.tecnm.mx (V.M.R.-V.); lucia.vc@tuxtla.tecnm.mx (L.M.C.V.-C.); francisco.rm@tuxtla.tecnm.mx (F.A.R.-M.); juan.vm@tuxtla.tecnm.mx (J.J.V.-M.); 2Departamento de Investigación y Desarrollo, 3R Biotec SA de CV, Tuxtla Gutiérrez 29000, Chiapas, Mexico; 3Departamento de Microbiología y Genética, Universidad de Salamanca, 37007 Salamanca, Spain; jdflores@usal.es

**Keywords:** biofertilizer, bioreactor prototype, cell growth rates, PGPB, *Sinorhizobium*

## Abstract

The implementation of bioreactor systems for the production of bacterial inoculants as biofertilizers has become very important in recent decades. However, it is essential to know the bacterial growth optimal conditions to optimize the production and efficiency of bioinoculants. The aim of this work was to identify the best nutriment and mixing conditions to improve the specific cell growth rates (µ) of two PGPB (plant growth-promoting bacteria) rhizobial strains at the bioreactor level. For this purpose, the strains *Sinorhizobium mexicanum* ITTG-R7^T^ and *Sinorhizobium chiapanecum* ITTG-S70^T^ were previously reactivated in a PY-Ca^2+^ (peptone casein, yeast extract, and calcium) culture medium. Afterward, a master cell bank (MCB) was made in order to maintain the viability and quality of the strains. The kinetic characterization of each bacterial strain was carried out in s shaken flask. Then, the effect of the carbon and nitrogen sources and mechanical agitation was evaluated through a factorial design and response surface methodology (RSM) for cell growth optimization, where µ was considered a response variable. The efficiency of biomass production was determined in a homemade bioreactor, taking into account the optimal conditions obtained during the experiment conducted at the shaken flask stage. In order to evaluate the biological quality of the product obtained in the bioreactor, the bacterial strains were inoculated in common bean (*Phaseolus vulgaris* var. Jamapa) plants under bioclimatic chamber conditions. The maximum cell growth rate in both PGPB strains was obtained using a Y-Ca^2+^ (yeast extract and calcium) medium and stirred at 200 and 300 rpm. Under these growth conditions, the *Sinorhizobium* strains exhibited a high nitrogen-fixing capacity, which had a significant (*p* < 0.05) impact on the growth of the test plants. The bioreactor system was found to be an efficient alternative for the large-scale production of PGPB rhizobial bacteria, which are intended for use as biofertilizers in agriculture.

## 1. Introduction

The continuous growth of the human population necessitates the production of an increasing amount of food to satisfy basic nutritional needs [1]. Consequently, there has been a significant rise in the systematic application of chemical fertilizers and other agro-inputs, such as fungicides, herbicides, and pesticides, to meet these societal demands. However, the indiscriminate use of toxic agrochemicals has resulted in severe disruptions to ecosystems and significant environmental problems, including soil degradation, erosion, poor soil health, and fertility loss, as well as water and air pollution [2]. As an emerging alternative, the use of biofertilizers formulated with plant growth-promoting bacteria (PGPBs) in agricultural systems has become a viable option to achieve sustainable ecological development and enhance crop fertility and productivity. Notably, the genera *Rhizobium*, *Sinorhizobium*, *Mesorhizobium*, *Bradyrhizobium*, *Devosia*, *Methylobacterium*, *Burkholderia*, and *Cupriavidus* stand out among the most beneficial microorganisms commonly utilized in biofertilizer formulations [3,4,5]. PGPB not only possesses nitrogen-fixing capabilities but also exhibits additional functional qualities that contribute to plant growth, including the production of phytohormones, siderophores, vitamins, and antibiotics [6,7].

Previous studies have primarily focused on the isolation and genomic analysis of native beneficial bacteria, leading to the identification of new rhizobial species [8]. The strains *Sinorhizobium mexicanum* ITTG-R7^T^ and *Sinorhizobium chiapanecum* ITTG-S70^T^ were isolated from nodules of the *Acaciella angustissima* shrubby legume [9,10]. These bacteria establish symbiotic relationships and form nodules in various legume species [10]. In field-level biofertilization experiments, these bacterial strains have demonstrated a positive impact on growth and fruit production in crops, as well as in other non-leguminous plant species [11]. These native bacteria exhibit desirable characteristics as plant growth-promoting (PGP) bacteria, including a high nitrogen fixation potential, phosphate solubilization, auxin synthesis, and siderophore production [12,13,14].

A crucial aspect of biofertilizer production is the cultivation and growth of microorganisms [15]. Typically, the process begins at the laboratory level using shake flasks and is later expanded to pilot-scale bioreactors for bacterial production. These processes necessitate the careful consideration of factors related to bacterial growth and the use of a specific culture medium [16]. However, as biofertilizer production progresses to the pilot bioreactor level or larger scales, costs significantly increase, requiring advanced technical infrastructure [17]. Biofertilizer biomass production liquid fermentation is most commonly used since this system is highly homogeneous and allows a rough control of critical process variables, such as temperature, pH, dissolved oxygen concentration, and agitation [18]. There are research works that are focused on the production of biofertilizers in bioreactors; these investigations have been developed in commercial brand bioreactors allowing for controlled conditions of temperature, pH, input and output flows, and sterility, as well as the measurement of variables, such as carbon dioxide concentration and oxygen [19,20]. Data regarding the large-scale production of *Rhizobium/Sinorhizobium* bacteria are scarce, and few studies have reported the production of these as biofertilizers in bioreactors. Ruiz-Valdiviezo et al. [21] reported the production of biofertilizers formulated with *Rhizobium calliandrae* LBP2-1^T^ and *S. mexicanum* ITTG R7^T^ in shaken flasks. These authors found that the carbon source and agitation played a crucial role in biomass production, resulting in concentrations above 10^9^ CFU/mL. Furthermore, after 240 days of storage, both rhizobial strains exhibited a significant positive impact on plant growth, nodulation, and the total N content in inoculated common bean plants. On the other hand, Trujillo-Roldan et al. [22] scaled up a shake flask process to a 10 L laboratory-scale bioreactor and a 1000 L pilot-scale bioreactor for the production of the PGP bacterium, *Azospirillum brasilense,* in a liquid inoculant formulation. They reported a concentration of 3.5 to 7.5 × 10^8^ CFU/mL obtained in shake flasks and the bioreactor. Interestingly, the bacterial inoculants had a positive impact on the maize crop’s yield even after they were stored for two years at room temperature. In this case, the optimal conditions for achieving the best growth and biomass production of the *Azospirillum* strain were found to be a volumetric mass transfer coefficient (KLa) of 31 h^−1^ and an agitation speed of 200 rpm. Furthermore, Gamboa-Suasnavart et al. [23] conducted a scale-up process from shake flasks to a bioreactor, taking into account the power input and the morphology of *Streptomyces lividans* to produce recombinant APA (45/47 kDa protein) from *Mycobacterium tuberculosis*. These researchers observed that the culture conditions in shake flasks influenced the morphology of filamentous *S. lividans*, as well as the productivity and O-mannosylation of the Ala-Pro-rich recombinant O-glycoprotein from *M. tuberculosis*. However, it should be noted that these studies highlight high equipment costs and complex operating conditions, which limit their applicability for research purposes or their adoption in agricultural sectors in rural areas.

In recent years, there has been a growing interest among local farmers in utilizing these native bacteria as biofertilizers for application in diverse strategic food crops in Mexico [24]. In response to the increasing demand for biofertilizers, local producers have resorted to the manual production of microbial inoculants. Due to limited knowledge and inadequate infrastructure, a wide array of products have been generated, comprising vermicompost, leachate, and fertilizers derived from livestock and poultry manure. Unfortunately, these products harbor a substantial bacterial pathogen load, which presents significant risks to human health and leads to considerable alterations in the soil microbiota [25,26]. Therefore, local farmers in Chiapas, Mexico, have initiated programs aimed at independently producing their own biofertilizers using artisanal methods, with the goal of achieving high nutritional quality and contaminant-free food production. However, they face significant challenges, as they lack detailed knowledge regarding the production and formulation of microbial inoculants. The objective of this study was to identify the best nutrimental and mixing conditions from shake flasks to a homemade laboratory bioreactor to achieve the efficient biomass production of native PGP bacterial strains *S. mexicanum* ITTG-R7^T^ and *S. chiapanecum* ITTG-S70^T^.

## 2. Materials and Methods

### 2.1. Bacterial Strains

The native nitrogen-fixing strains *S. mexicanum* ITTG-R7^T^ (DQ411930) and *S*. *chiapanecum* ITTG-S70^T^ (EF457949) were employed in this study. These strains are novel *Sinorhizobium* species isolated from the shrubby legume *Acaciella angustissima* [10]. They are recognized for their capacity as plant growth-promoting bacteria (PGPB). Furthermore, they exhibit non-toxic and non-pathogenic characteristics. Dr. Reiner Rincón Rosales (from Tecnologico Nacional de México) generously provided these strains, which are preserved in the culture collection of the Genomic Ecology Laboratory at Tecnologico de Tuxtla Gutierrez, Chiapas, Mexico.

### 2.2. Initial Characterization of Bacterial Growth

The bacterial strains ITTG-R7^T^ and ITTG-S70^T^ were precultured in 500 mL Erlenmeyer flasks at 30 °C and 150 rpm, using 100 mL of a PY-Ca^2+^ culture medium. The PY-Ca^2+^ medium composition per liter included 5.0 g of casein peptone, 10 mL of 0.94 M CaCl_2_, and 3.0 g of yeast extract [11]. Subsequently, the strains were characterized for growth kinetics in 250 mL Erlenmeyer flasks containing 50 mL of the PY-Ca^2+^ medium at 30 °C and 120 rpm. Growth was measured by optical density (OD) at 600 nm using a Beckman Coulter DU730 spectrophotometer (Beckman Coulter, Inc, California, USA). Biomass production was determined based on fresh weight during bacterial growth. At the initial time (T_0_), 1 mL of inoculated culture medium was poured into a pre-weighed 1.5 mL Eppendorf tube (Eppendorf, Hamburg, Germany). The sample was centrifuged at 10,000 rpm for 1 min. The supernatant was discarded, and the difference in weight between the empty tube and the tube with the sample provided the weight of the fresh biomass. This process was performed in triplicate at different sampling times until the stationary phase. To maintain the stability and viability of the strains, a masterwork cell bank was created following the methodology suggested by Del Puerto et al. [27]. The bacteria were preserved in a mixture of 30% glycerol and 70% PY-Ca^2+^ broth and stored at −20 °C until use.

### 2.3. Experimental Design for Optimizing Native Sinorhizobium Strain Growth Conditions

A factorial design and response surface methodology (RSM) were used to identify the optimal nutritional and mixing conditions for *S. mexicanum* ITTG-R7^T^ and *S. chiapanecum* ITTG-S70^T^. This experimental design employed a 3^2^-factorial design. Two independent variables, namely culture medium (X_1_) and stirring (X_2_), were evaluated. The culture media used as test levels were as follows: Y-Ca^2+^ containing per liter: [10 mL of 0.94 M CaCl_2_ and 8.0 g of yeast extract at pH 6.8]; PY-Ca^2+^: [5.0 g of casein peptone, 10 mL of 0.94 M CaCl_2_, and 3.0 g of yeast extract at pH 6.8]; YEM: [10 g of mannitol, 0.5 g of K_2_HPO_4_, 0.2 g of MgSO_4_, 0.1 g of NaCl, 3.0 g of CaCO_3_, and 3.0 g of yeast extract at pH 6.8]. Each independent variable had three levels: a low level (−1), an intermediate level (0), and a high level (+1) (Table 1). Nine treatments in triplicate were evaluated, resulting in a total of 27 experimental units for each bacterial strain. The experimental unit consisted of a 250 mL Erlenmeyer flask containing 50 mL of the culture medium. During the cultivation of the strains, pH was maintained at 6.8, and the temperature was set at 30 °C in accordance with the recommendations suggested by Berovic [28].

The response function (*y*) measured was the specific growth rate (*μ*). This value was related to the coded variables (X_i_, i = 1 and 2) by a second-degree polynomial. The polynomial equation proposed for the two responses (*y*) was:(1)$$y=b_{0}-b_{1}x_{1}+b_{2}x_{2}+b_{1}x_{1}^{2}-b_{12}x_{1}x_{2}+b_{2}x_{2}^{2}$$
where *y* is the predicted response for the specific growth rate (*μ*), *b*_0_ = constant, *b*_1_ = culture medium, and *b*_2_ = stirring; *b*_1_ and *b*_2_ = linear coefficients; *b*_11_, and *b*_22_ = quadratic coefficients, and b_12_ = cross-product coefficients.

For the measurement of (*μ*), the OD was determined by UV-visible spectrophotometry at 600 nm. Data obtained in the experimental assays were analyzed by an analysis of variance ANOVA (*p* < 0.05) in order to determine differences between the treatments. The effect and regression coefficients of the individual linear, quadratic, and interaction terms were determined. The regression models were used to generate response surface plots.

### 2.4. Design and Construction of a Homemade Bioreactor

Based on the bacterial growth conditions determined during the optimization test at the flask level, a homemade bioreactor was designed and constructed. The bioreactor utilized a wide-mouth glass jar with a nominal capacity of 3.8 L (“6.10” diameter and “9.95” height). The design of various components, including the lid, clamp, motor base, aeration system, impeller, impeller shaft, baffles, and rotation system, was developed using Solidworks software (BIOVIA, Dassault Systèmes, Dearborn, MI, USA). The impeller chosen was of the Rushton type, with a diameter equal to 1/3 of the tank’s diameter, and four baffles were incorporated, with each baffle width being 1/12 of the tank’s diameter. The complete prototype design is currently undergoing a utility model registration process with the Mexican Institute of Industrial Property (IMPI) under registration number MX/u/2023/000016 (https://www.gob.mx/impi) (accessed on 16 January 2023).

### 2.5. Bacterial Growth Assay in Homemade Bioreactor

Initially, a sterilization test was conducted on the homemade bioreactor to assess its construction quality and ensure aseptic conditions for the cultivation of rhizobial bacteria. The equipment was autoclaved at 15 psi and 121 °C for 20 min. The cell biomass production efficiency in the homemade bioreactor was determined by fresh weight, considering an operating volume of 2.6 L for the Y-Ca^2+^ culture medium at 200 rpm and 0.3 vvm. During the cultivation of the strains, the pH was maintained at 6.8, and the temperature was 30 °C. At the initial time point (T_0_), 1 mL of the inoculated culture medium was transferred into a pre-weighed 1.5 mL Eppendorf tube. The sample was centrifuged at 10,000 rpm for 1 min, and the supernatant was discarded. The weight difference between the empty tube and the tube with the sample provided the fresh biomass weight. This process was carried out in triplicate at different sampling times until the stationary phase was reached.

A prediction model for biomass production in the homemade bioreactor was developed using the least squares method. The non-linear GRG algorithm [29] was employed to obtain a polynomial equation for bacterial growth.

### 2.6. Bacterial Effectiveness through Plant Inoculation Assay

In order to assess the effectiveness of the bacterial strains produced in the homemade bioreactor, an inoculation test was conducted on common bean plants (*Phaseolus vulgaris* L.) cv. Jamapa (Appendix A). The seeds were subjected to a disinfection process involving a 5 min treatment with 70% ethanol, followed by a 15 min treatment with 25% sodium hypochlorite, and subsequent rinsing with sterile water [10]. The seeds were then germinated on plates containing an 0.8% water–agar medium until the radicle emerged. Erlenmeyer flasks containing vermiculite moistened with an N-free Fahraeus solution [0.1 g L^−1^ CaCl_2_; 0.12 g L^−1^ MgSO_4_·7H_2_O; 0.1 g L^−1^ KH_2_PO_4_; 0.15 g L^−1^ Na_2_HPO_4_·2H_2_O; 0.005 g L^−1^ Fe citrate; and trace amounts of Mn, Cu, Zn, B, Mo; pH 6.8] were used to grow the plants [30]. The inoculation assay employed a completely randomized design, evaluating four treatments with six replicates each. The first treatment (T_1_) and second treatment (T_2_) consisted of *S. mexicanum* ITTG-R7^T^ and *S. chiapanecum* ITTG-S70^T^, respectively. The inoculants were obtained from the 3.8 L homemade bioreactor, cultivated in a Y-Ca^2+^ culture medium at 26 °C, and adjusted to a final concentration of 10^6^ CFU ml^−1^. The third treatment (T_3_) involved plants treated with the triple 17 fertilizer (17% N, 17% P, and 17% K) as the positive control. The fourth treatment (T_4_) served as a negative control, consisting of non-inoculated plants. The plants were cultivated in controlled plant chamber conditions. After a 60-day period of cultivation, the plants were meticulously collected, and the following variables were determined: the plant’s total height (cm), dry weight (g), root weight (g), the number of nodules per plant, the total chlorophyll content, and total nitrogen content (mg per dry plant). All the variables were measured following the methodology described by Rincón-Molina et al. [31].

### 2.7. Statistical Analysis

The data obtained from the optimization of bacterial growth conditions and the inoculation test were analyzed using ANOVA at a significance level of alpha = 0.05, using the statistical software Statgraphics Centurion XV.2 (The Plains, VA, USA). Mean comparisons were conducted using the Tukey test (*p* < 0.05). Additionally, the main effects plots and response surface plots with contour areas were generated using Minitab V.16 software (State College, PA, USA).

## 3. Results

### 3.1. Bacterial Growth Characteristics

The specific growth rate (*µ*) of strain *S. mexicanum* ITTG-R7^T^ was 0.1465 ± (0.02) h^−1^, and for strain *S. chiapanecum* ITTG-S70^T^ was 0.1367 ± (0.01) h^−1^. In both, it was observed that the transition phase between the growth and the stationary phases occurred after 12 h of culture time (Figure 1). Biomass production before the stationary phase was 12.43 ± (1.26) g L^−1^ and 8.32 ± (0.50) g L^−1^, respectively. Additionally, for both strains, diauxic growth was observed during the log phase. 

### 3.2. Optimization of the Growth of Native Sinorhizobium Strains

The analysis of variance (ANOVA) of the growth rates revealed that factors X_1_ (culture medium) and X_2_ (stirring) had the most significant effect on the maximum growth rate (*µ*) of strain ITTG-R7^T^, with *p*-values < 0.0000 and <0.0002, respectively (Table 2). Furthermore, it was found that factor X_1_ (culture medium) had a significant effect (*p* < 0.0190) on the growth rate of strain ITTG-S70^T^, whereas factor X_2_ (stirring) did not exhibit a significant effect on µ. Furthermore, an interaction effect was observed between factors X_1_ (culture medium) and X_2_ (stirring), which significantly influenced (*p* < 0.0003) the growth of the strain ITTG-R7 ^T^.

The Pareto charts allowed us to determine the standardized effects of the evaluated factors on the maximum growth rate (Figure 2). In the case of strain ITTG R7^T^, it was evident that the factors, culture medium (X_1_) and agitation (X_2_), exerted a significant influence (*p* < 0.05) on the growth rate. Furthermore, the interaction [culture medium *X* agitation] had an effect on the variable under study (Figure 2A). For strain ITTG S70^T^, only agitation had an impact on the growth of this bacterial species (Figure 2B). Both graphs indicate that the culture medium had a negative influence on the range of bacterial growth, while agitation had a positive effect.

The main effects plots provided a visual representation of the effects of factors X_1_ (culture medium) and X_2_ (stirring) on the growth of the bacterial strains (Figure 3). *S. mexicanum* ITTG R7^T^ exhibited its highest growth rate when cultivated in the Y-Ca^2+^ medium (level −1.0) with a mechanical agitation of 300 rpm (level 1.0) (Figure 3A). On the other hand, the ITTG S70^T^ strain achieved its maximum growth rate (*µ*) when cultured in the Y-Ca^2+^ medium (level −1.0) but with an agitation of 200 rpm (Figure 3B).

Table 3 presents the growth range (*µ*) of the strains *S. mexicanum* ITTG R7^T^ and *S. chiapanecum* ITTG S70^T^ under optimal culture medium and shaking conditions. The ITTG R7^T^ strain exhibited the highest growth rate [*µ* = 0.7324 ± (0.030) h^−1^] when cultivated in the Y-Ca^2+^ medium at 300 rpm. Similarly, the ITTG S70^T^ strain demonstrated a greater optimal growth range (*µ* = 0.3830 ± (0.054) h^−1^) when the Y-Ca^2+^ culture medium was utilized as a carbon and nitrogen source and maintained under constant stirring at 300 rpm. By contrast, both *Sinorhizobium* strains displayed a low growth rate (*µ*) when cultivated in the YEM medium at an agitation of 120 rpm.

The optimal culture conditions were determined by fitting the experimental data to a regression model that incorporated the variables X_1_ (culture medium) and X_2_ (stir). The equations for optimizing the response variable (*µ*) of the strains *S. mexicanum* ITTG-R7^T^ and *S. chiapanecum* ITTG-S70^T^ were obtained under optimal culture conditions, which involved using a culture medium (CM) and stirring (*S*) while maintaining a constant temperature of 30 °C and a pH of 6.8.

For the strain *S. mexicanum* ITTG-R7^T^, the growth optimization equation was:(2)$$\mu_{Max}=0.283-0.102\left(CM\right)+0.086\left(S\right)+0.040\left(CM^{2}\right)-0.100\left(CM\right)\left(S\right)+0.041\left(S^{2}\right)$$
and for strain *S. chiapanecum* ITTG-S70^T^’, the growth optimization equation was:(3)$$\mu_{Max}=0.331-0.032\left(CM\right)+0.023\left(S\right)+0.029\left(CM^{2}\right)-0.021\left(CM\right)\left(S\right)+0.037\left(S^{2}\right)$$

The surface plot describes the behavior of the maximum growth range (*µ*) across the experimental region (Figure 4). Notably, the points where the surface reaches higher values correspond precisely to optimal treatments or conditions where the *Sinorhizobium* strains exhibit robust growth. Examining Figure 4A, we observed that the response surface for the ITTG R7^T^ strain indicated its favorable growth when the culture medium was at a lower level (−1), specifically the Y-Ca^2+^ medium, and agitation was at a higher level (+1), which corresponded to 300 rpm. Similarly, from Figure 4B, we can determine that the optimal growth range for the ITTG S70^T^ strain occurred when it was cultivated in the Y-Ca^2+^ medium at a lower intermediate level (0) of agitation, corresponding to 200 rpm. Therefore, it is evident that the culture medium played a critical role in achieving the optimal growth of the native bacterial strains while agitation could be adjusted within the range of 200 to 300 rpm.

### 3.3. Production Efficiency in Homemade Bioreactor

A homemade stirred-tank bioreactor was fabricated from the reconversion of a borosilicate glass jar with a 3.8 L capacity (nominal volume). The lid, clamp, and motor base were 3D-printed in Phrozen TR300 resin. The aeration system, including the Rushton impeller, the impeller shaft, baffles, and the rotation system, were made of stainless steel (Figure 5). Sterilization tests performed on the assembled system and loaded with a culture medium showed that it maintained sterile conditions during the evaluation period (5 days).

To analyze the biomass production efficiency of the 3.8 L homemade bioreactor, the growth kinetics of the bacterial strains *S. mexicanum* ITTG-R7^T^ (Figure 6A) and *S. chiapanecum* ITTG-S70^T^ (Figure 6B) were obtained following the optimal conditions previously determined in the main effects plots. A maximum sampling time of 12 h was chosen for both growth kinetics in order to ensure consistency and standardization in the sample collection. A production of 18.53 ± (1.97) g L^−1^ and a biomass productivity of 1.54 ± (0.16) g L^−1^ h^−1^ for ITTG-R7^T^ were reached after 12 h of growth in the Y-Ca^2+^ medium and 300 rpm. By contrast, a biomass concentration of 18.67 ± 0.40 g L^−1^ and a specific growth rate of 1.56 ± 0.04 g L^−1^ h^−1^ were achieved after 12 h of growth in the Y-Ca^2+^ medium at 200 rpm for ITTG-S70^T^. The prediction model that accurately described biomass production (*BIOM*) as a function of culture time (*x*) had an R^2^ value of 1.0.

For the strain *S. mexicanum* ITTG-R7^T^ (Figure 6A):(4)$$BIOM=-0.0078x^{3}+0.2254x^{2}-0.3705x+4.1103$$
and, for strain *S. chiapanecum* ITTG-S70^T^ (Figure 6B),
(5)$$BIOM=-0.0113x^{3}+0.2479x^{2}-0.2594x+5.7439$$

The prediction models for biomass production demonstrated a perfect fit with an R^2^ value of 1.0. This high level of accuracy confirmed the reliability of the models in estimating biomass production based on culture time. These results emphasized the significance of the prediction model in accurately predicting biomass production for the studied native *Sinorhizobium* strains.

In terms of the biomass production yield under optimal growth conditions, significant differences (*p* < 0.05) were observed among the indigenous bacterial strains when different culture media were employed (Table 4). The strain *S. mexicanum* ITTG-R7^T^ exhibited higher biomass production (18.53 ± 1.97 g L^−1^) when cultivated in the Y-Ca^2+^ medium, while a lower biomass yield (12.43 ± 1.26 g L^−1^) was obtained in the PY-Ca^2+^ medium, representing a 49.0% variation between the culture media. Similarly, the strain *S. chiapanecum* ITTG-S70^T^ demonstrated a substantial biomass accumulation (18.67 ± 0.40 g L^−1^) when grown in the Y-Ca^2+^ medium, resulting in a 124.4% difference in biomass production compared to its performance in the PY-Ca^2+^ medium. These results underscore the significance of selecting the appropriate culture medium to attain an optimal biomass yield in native Sinorhizobium strains. The Y-Ca^2+^ medium proved to be more conducive to biomass production, highlighting the potential for enhancing growth and productivity by optimizing the growth conditions.

On the other hand, the cost analysis (Table 5) of the culture media used for the biomass production of strains *S. mexicanum* ITTG R7^T^ and *S. chiapanecum* ITTG-S70^T^ determined that the Y-Ca^2+^ culture medium was 49% more cost-effective compared to the YEM medium, and 36% more cost-effective compared to the PY-Ca^2+^ medium per liter of the culture medium formulated for the homemade bioreactor.

### 3.4. Efficacy of Native Sinorhizobium Strains in Promoting Plant Growth

The results of the inoculation test using the *S. mexicanum* ITTG-R7^T^ and *S. chiapanecum* ITTG-S70^T^ strains cultured in the homemade bioreactor demonstrated the retention of their biological qualities, including high infectivity and effectiveness as plant growth-promoting bacteria (Table 6). The ITTG-R7^T^ strain exhibited superior effects on various growth parameters, including the total height, total weight, root weight, the number of nodules, and total nitrogen content (*p* < 0.05), outperforming both the negative control (non-inoculated plants) and plants treated with an NPK triple 17 fertilizer. Additionally, the ITTG-S70^T^ strain significantly increased chlorophyll levels (3.2 mg g^−1^) compared to other treatments (*p* < 0.05). Both strains effectively induced nodulation, with ITTG-R7^T^ demonstrating the highest number of nodules per plant (51 nodules). These findings confirm the potential of native *Sinorhizobium* strains as biofertilizers, maintaining their biological attributes, infectivity, and plant growth-promoting capabilities when cultivated in the homemade bioreactor.

## 4. Discussion

In recent years, the demand for eco-friendly agricultural practices has increased to mitigate the environmental impact of chemical fertilizers [32]. Sustainable agriculture necessitates strategies that enhance food production safety, landscape quality, and fertility conditions while gaining social acceptance [33]. Beneficial microorganisms as biofertilizers offer a sustainable and environmentally friendly approach to optimizing crop growth and productivity. However, scaling up biofertilizer production presents challenges, including attaining high cell concentrations, ensuring product safety by eliminating contaminants, and reducing production costs [34]. Overcoming these challenges is crucial for the widespread adoption of biofertilizers as efficient and viable solutions in modern agriculture [35].

This study presents, for the first time, the growth kinetics, optimal conditions, and production of native *Sinorhizobium* strains using a novel homemade bioreactor system. These strains exhibited rapid growth and high exopolysaccharide production. The bioreactor, designed for the *Sinorhizobium* bacteria, enabled high cell concentrations and efficient biofertilizer production. This approach provides a scalable method for producing larger quantities of bioinoculants compared to flask-level systems. These results could contribute to the development of sustainable, cost-effective, and user-friendly biofertilizer production methods.

According to the response surface methodology, it was determined that the Y-Ca^2+^ culture medium and 200 rpm stirring rate were the most suitable for the growth of the native *Sinorhizobium* strains. These results indicate that the addition of the yeast extract and soluble Ca^2+^ provided the essential nutrients for the growth of this bacterial species. Techniques for the isolation and cultivation of nitrogen-fixing bacteria have been reported by different authors [36,37,38]. The most common culture medium for growing bacterial species within the genus *Sinorhizobium* is YEM, which contains mannitol as a carbon source [39,40], and PY, which contains casein peptone [41,42]. Here, we report that both strains are able to grow in a culture medium with CaCl_2_ and yeast extract. This result completely differs from the works mentioned above, where the authors proposed cultural media that were complex in composition. The yeast extract is the most commonly used component in media to produce microbial biomass [43,44,45], and it is the main component in the culture medium presented in this work. The chemical components of the yeast extract are ill-defined due to extreme variations between the raw materials used in manufacturing processes. However, it is broadly known that the composition of the yeast extract mainly consisted of amino acids, peptides, growth factors (vitamins like thiamine, riboflavin, pantothenic acid, pyridoxine, niacin, cyanocobalamin, and other hydrosoluble vitamins of the B complex), trace elements, carbohydrates, and others [46,47]. The concentration of the yeast extract tested in this study allowed us to assume that this raw material acted as a source of both nitrogen and growth factors; this statement matches with the scientific reports that mention the exigent vitamin requirements in rhizobial strains [36,48]. Toledo et al. [42] demonstrated the successful isolation and cultivation of the *Sinorhizobium americanum* strain using the PY medium, which served as a specific and effective culture medium. Similarly, Rincón-Rosales et al. [8] achieved pure isolates of native *Rhizobium* strains with a high nitrogen fixation capacity using this identical culture medium. These studies highlight the reliability and efficiency of the PY medium for the isolation and cultivation of the *Sinorhizobium* and *Rhizobium* strains, respectively. In addition, it has been reported that rhizobial strains have various metabolic strategies to process nitrogen, highlighting the ubiquitous glutamine synthetase (GS) enzyme system for obtaining energy, and promoting the synthesis of important biomolecules from yeast extract glutamic acid [49]. This allowed us to realize the ability of these strains to grow in a culture medium with a nitrogen source that was rich in amino acids without requiring a carbon source such as mannitol.

On the other hand, the addition of CaCl_2_ to the culture medium (Y-Ca^2+^) played a key role in preventing toxicity from some components present in the yeast extract. Divalent cations, especially Ca^2+^, have been shown to prevent toxicity caused mainly by glycine and sodium chloride [50]. Additionally, the addition of CaCl_2_ improves the protein channels of exclusion in the bacterial cell membrane and influences the production of metabolites, such as EPS, that contribute to alleviating toxicity due to high salinity concentrations.

The findings in this work contribute directly to obtaining bacterial biomass using a culture medium with components that are easily accessible in raw materials, the preparation of which is simple and fast. Additionally, we demonstrated that when culturing *Sinorhizobum* strains, it is possible to exclude the most expensive ingredient of YEM broth (mannitol); this modification allowed for an over 35% reduction in the cost with the modified medium (Y-Ca^2+^) compared to YEM and PY, as shown in Table 4. This highlights the significance of the Y-Ca^2+^ medium, not only for its efficiency in promoting growth and biomass production but also for its cost-effectiveness, especially in the context of a homemade bioreactor setup. Consequently, the use of the Y-Ca^2+^ medium presents an appealing and suitable option for the large-scale production of these rhizobial bacteria.

In this study, agitation played a critical role in the cultivation and production of rhizobial bacteria in a homemade bioreactor. Shaking at 200–300 rpm ensured the homogeneous mixing of the culture medium, providing sufficient nutrients and oxygen while facilitating waste product removal. Furthermore, optimal shaking conditions prevented cell aggregation, promoting bacterial growth and reproduction [16]. Therefore, the precise control of agitation is essential for maximizing bioreactor efficiency and bacterial cell production, as highlighted by previous studies [17,18]. In summary, proper agitation in the handmade bioreactor was essential for optimizing the growth and production of the native *Sinorhizobium* bacteria and ensuring the quality of the culture.

In addition, the inoculation of *Phaseolus vulgaris* plants with bacterial inoculants obtained from the homemade bioreactor confirmed the preserved biological functionality of the *Sinorhizobium* strains. It is crucial for the bacteria cultivated in the bioreactor to possess a high infectivity capacity, effectiveness, and cell viability, with a cell concentration exceeding 1 × 10^6^ CFU/mL to ensure favorable impacts on plant growth. These strains displayed remarkable infectivity, the ability to form nodules, and exhibited effective nitrogen fixation. Remarkably, the *Sinorhizobium* strains showcased the superior growth and development of bean plants compared to alternative treatments, owing to their plant growth-promoting attributes and adaptability to leguminous plants [10,14,32]. The establishment of symbiotically efficient root nodules by bacterial strains resulted in enhanced plant height, total weight, and root weight, facilitated by the legume–rhizobial symbiosis that enhances biological nitrogen fixation. These findings underscore the potential of cultivating and producing nitrogen-fixing *Sinorhizobium* bacteria at scale utilizing a homemade bioreactor and lay the groundwork for modeling analogous processes for other bacterial species with biofertilizer potential.

Obtaining bioinoculants with plant growth-promoting properties, especially nitrogen fixation, through cost-effective and user-friendly production systems provides an alternative for farmers to address environmental pollution and economic challenges associated with chemical fertilizers. In this regard, careful attention should be given to the selection of specific bacterial strains for formulating and producing bioinoculant products. These strains must not only enhance crop growth but also ensure safety for humans, plants, soil, and the environment. Farmers can use autoclaves and appropriate equipment to sterilize the bioreactor. Providing training on equipment handling and maintenance is crucial. Although homemade bioreactors are user-friendly, farmers may benefit from guidance in growth media preparation and bacterial culture management. Offering technical advice is recommended. The cost of home bioreactors is justified considering the long-term benefits of producing cost-effective biofertilizers, reducing reliance on expensive chemical fertilizers, and promoting sustainable farming. Home bioreactors represent an economically viable and environmentally friendly approach for farmers, especially in vulnerable agricultural sectors. The production process of biofertilizers in homemade bioreactors should be economically viable, facilitating the adoption of these biotechnologies by various vulnerable agricultural sectors, particularly in Mexico, which aim to reduce excessive synthetic fertilizer application and dependence by embracing new technologies.

## 5. Conclusions

Bacterial production in the laboratory prototype bioreactor led to appropriate cell concentrations. This is the first report on the use of a homemade, economical, and efficient bioreactor designed in southern Mexico for the production of rhizobial biofertilizers from native strains. Thus, this study introduces a new approach for obtaining larger amounts of inoculants, contributing to the overall biotechnological approaches for biofertilizer production for use in conventional agriculture systems and industries. The results demonstrate that the elite nitrogen-fixing bacterial strains *S. mexicanum* ITTG-R7^T^ and *S. chiapanecum* ITTG-S70^T^, cultivated in a Y-Ca^2+^ culture medium and at 200–300 rpm, were highly effective in promoting plant growth when used to inoculate bean plants. These findings have important implications for sustainable agriculture practices that aim to reduce reliance on synthetic fertilizers and promote the use of environmentally friendly options, such as bio-inoculants. The bioreactor system was found to be an efficient and low-cost alternative for the large-scale production of PGPB rhizobial bacteria intended for use as biofertilizers in agricultural crops.

## 6. Patents

The prototype bioreactor design was registered as a utility model with the Instituto Mexicano de la Propiedad Industrial (IMPI), registration number MX/u/2023/000016.

## Figures and Tables

**Figure 1 bioengineering-10-00960-f001:**
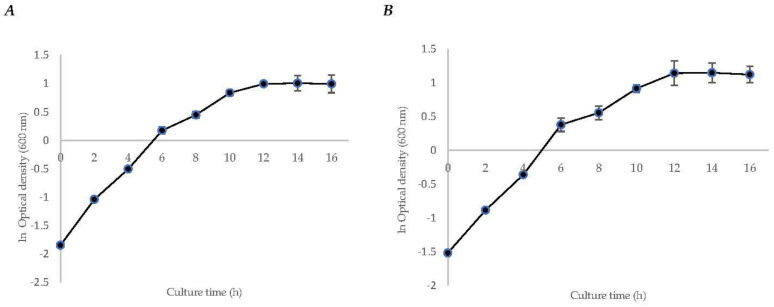
Initial characterization kinetics of bacteria growth for strains of (**A**) *S. mexicanum* ITTG-R7^T^ and (**B**) *S. chiapanecum* ITTG-S70^T^ grown in shaken flasks.

**Figure 2 bioengineering-10-00960-f002:**
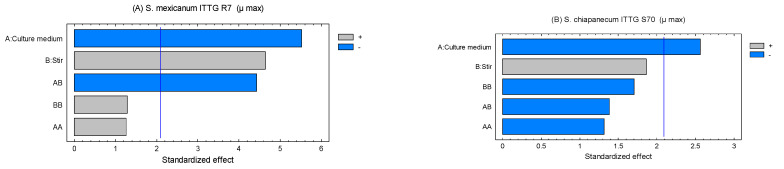
Pareto charts of standardized effects on culture medium and stirring on maximum growth rate (µ) in native *Sinorhizobium* bacteria. (**A**) *S. mexicanum* ITTG-R7^T^ and (**B**) *S. chiapanecum* ITTG-S70^T^.

**Figure 3 bioengineering-10-00960-f003:**
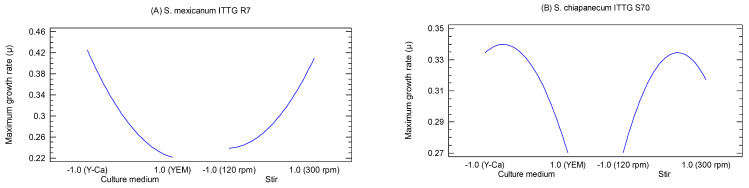
Main effects plots on bacterial maximum growth rate (*µ*). (**A**) *S. mexicanum* ITTG-R7^T^ and (**B**) ‘*S*. *chiapanecum* ITTG-S70^T^’.

**Figure 4 bioengineering-10-00960-f004:**
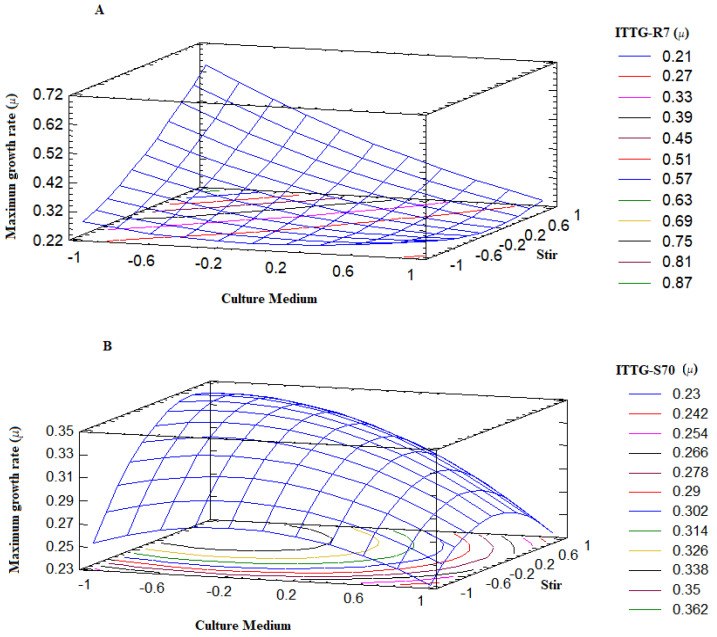
Response surface plots on bacterial maximum growth rate. (**A**) *S. mexicanum* ITTG-R7^T^ and (**B**) *S. chiapanecum* ITTG-S70^T^.

**Figure 5 bioengineering-10-00960-f005:**
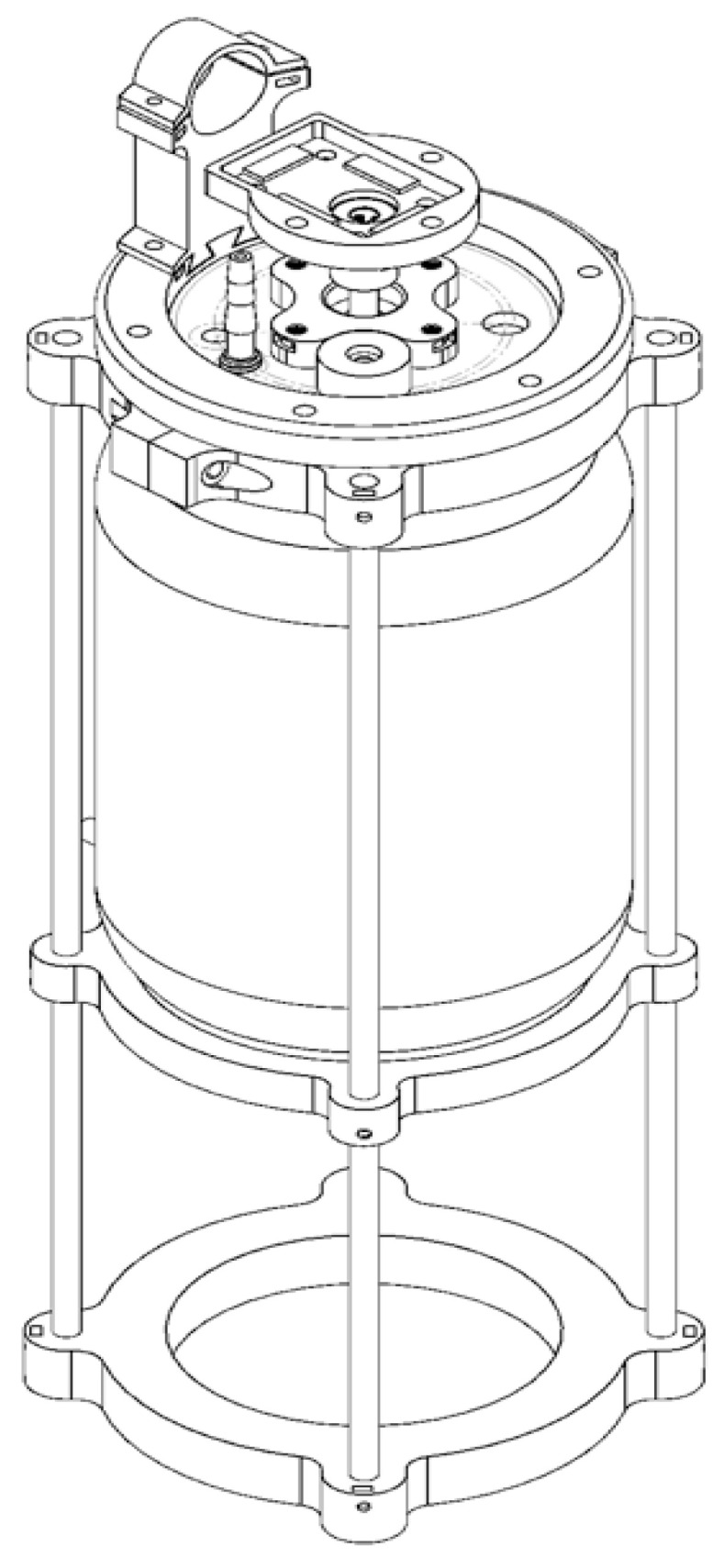
Prototype of a homemade stirred-tank bioreactor registered with the Mexican Institute of Industrial Property (IMPI), registration number MX/u/2023/000016 (https://www.gob.mx/impi) (accessed on 16 January 2023).

**Figure 6 bioengineering-10-00960-f006:**
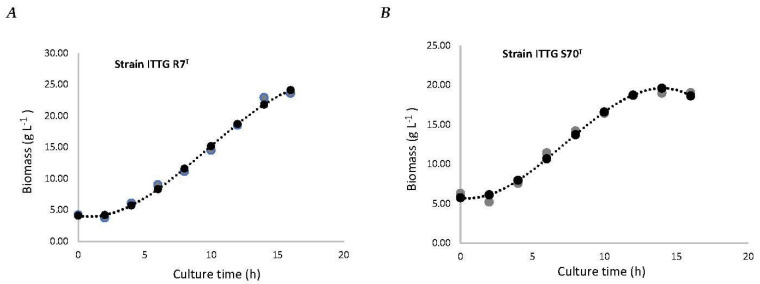
Biomass production of bacterial strains: (**A**) *S. mexicanum* ITTG-R7^T^ and (**B**) *S. chiapanecum* ITTG-S70^T^ in Y-Ca^2+^ medium. Grey dots represent the measurements of the experiment and black dots are the mathematical model adjustment.

**Table 1 bioengineering-10-00960-t001:** Experimental design employed for optimization of bacterial growth conditions.

Factors	Levels		
	Low (−1)	Intermediate (0)	High (+1)
X_1_: Culture medium	Y-Ca^2+^	PY-Ca^2+^	YEM
X_2_: Stirring (rpm)	120	200	300

**Table 2 bioengineering-10-00960-t002:** Significance level (*p*-value) of optimal growth conditions for bacterial strains.

Variables	*S. mexicanum* ITTG-R7^T^ *p*-Value *	*S. chiapanecum* ITTG-S70^T^ *p*-Value
x_1_: culture medium	0.0000 **	0.0190 *
x_2_: stirring	0.0002 **	0.0778 ^NS^
x_1_x_1_	0.2238 ^NS^	0.2034 ^NS^
x_1_x_2_	0.0003 **	0.1814 ^NS^
x_2_x_2_	0.2126 ^NS^	0.1038 ^NS^
error	0.8871 ^NS^	0.7766 ^NS^

* Statistically significant (*p* < 0.05), ** highly significant, ^NS^ not significant.

**Table 3 bioengineering-10-00960-t003:** Growth ranges (µ) achieved by indigenous *Sinorhizobium* strains in the optimization experiments using an experimentally calculated optical density (OD).

Strain	Culture Medium	Stirring (rpm)		
		120	200	300
*S. mexicanum* ITTG-R7^T^	Y-Ca^2+^	0.3151 ± (0.037) h^−1^	0.3113 ± (0.021) h^−1^	0.7324 ± (0.030) h^−1^
	PY-Ca^2+^	0.2460 ± (0.002) h^−1^	0.3595 ± (0.014) h^−1^	0.3267 ± (0.005) h^−1^
	YEM	0.2352 ± (0.008) h^−1^	0.2595 ± (0.004) h^−1^	0.2519 ± (0.035) h^−1^
*S. chiapanecum* ITTG-S70^T^	Y-Ca^2+^	0.2897 ^⌘^ ± (0.013) ^¥^ h^−1^	0.2560 ± (0.008) h^−1^	0.3830 ± (0.054) h^−1^
	PY-Ca^2+^	0.2654 ± (0.005) h^−1^	0.3683 ± (0.005) h^−1^	0.3044 ± (0.017) h^−1^
	YEM	0.1985 ± (0.013) h^−1^	0.3308 ± (0.005) h^−1^	0.2065 ± (0.011) h^−1^

^⌘^ Bacterial growth rate (*µ*) expressed in h^−1^. ^¥^ Mean values of three replicates. In parentheses, standard deviation.

**Table 4 bioengineering-10-00960-t004:** Biomass production yield for *S. mexicanum* ITTG-R7^T^ and *S. chiapanecum* ITTG-S70^T^ in the optimal and reference culture media.

Culture Medium	Biomass Production g L^−1^	
	ITTG R7^T^	ITTG S70^T^
PY-Ca^2+^ (Reference medium)	12.43 ± (1.26)B ^¥^	8.32 ± (0.50)B
Y-Ca^2+^ (Optimal medium)	18.53 ± (1.97)A	18.67 ± (0.40)A
Variation (%)	49.0	124.4
*p* < 0.05	0.0106736	0.0000096

^¥^ Mean values of three replicates. Means followed by the same letter are non-significant according to Student’s *t*-test (*p* < 0.05).

**Table 5 bioengineering-10-00960-t005:** Cost analysis of PY-Ca^2+^, YEM, and modified media (Y-Ca^2+^) for *S. mexicanum* ITTG-R7^T^ and *S. chiapanecum* ITTG-S70^T^ cultures.

Ingredients	Cost (USD per Liter)		
	Y-Ca^2+^	YEM	PY-Ca^2+^
CaCl_2_	0.00163		0.00163
Yeast extract	1.02972	0.61939	0.61939
Casein peptone			1.05879
Mannitol		0.89997	
K_2_HPO_4_		0.02117	
MgSO_4_		0.01165	
NaCl		0.00159	
CaCO_3_		0.57174	
H_2_O	0.10588	0.10588	0.10588
Total	1.13723	2.23139	1.78569
Reduction		−49%	−36%

**Table 6 bioengineering-10-00960-t006:** Growth parameters for *P. vulgaris* plants inoculated with *S. mexicanum* ITTG-R7^T^ and *S. chiapanecum* ITTG-S70^T^.

Treatment	Total Height (cm)	Total Weight (g)	Root Weight (g)	Nodules Number	Total N (%)	Chlorophyll (mg g^−1^)
T_1_: *S. mexicanum* ITTG-R7^T^	61.25 A ^¥^	6.26 A	1.50 A	51 A	6.83 A	2.6 AB
T_2_: *S. chiapanecum* ITTG-S70^T^	45.75 B	3.25 B	0.90 B	32 B	4.07 BC	3.2 A
T_3_: NPK triple 17	37.25 C	2.62 C	0.66 BC	0 C	5.67 AB	1.5 B
T_4_: Negative control (non-inoculated, non-fertilized)	26.50 D	2.32 C	0.48 C	0 C	3.31 C	1.6 B
*p*-value	0.0000	0.0000	0.0000	0.0000	0.0009	0.0035
HSD ^£^ (*p* < 0.05)	6.7242	0.3510	0.2747	14.444	1.777	0.9604

^¥^ Mean values of six replicates. Means followed by the same letter are non-significant (Tukey test, *p* < 0.05). ^£^ HSD, honestly significant difference.

## Data Availability

The authors declare that all relevant data supporting the findings of this study are included in the article.

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
