# Peer review of "Cost-Effective Cultivation of Native PGPB Sinorhizobium Strains in a Homemade Bioreactor for Enhanced Plant Growth"

_bioengineering, 2023, doi:10.3390/bioengineering10080960_

Round 1
Reviewer 1 Report
The research manuscript titled" Cost-Effective Cultivation of Native PGPB Sinorhizobium Strains in a Homemade Bioreactor for Enhanced Plant Growth" was targeted at the agricultural field. There are some ambiguous that should be considered. After revision, this manuscript can be accepted.
Please answer the following comments in detail:
1. All abbreviations should be defined at first appearance in the Abstract. Two terms are not defined in the Abstract, such as PY-Ca2+ and PGPB. Following this revision, kindly modify "PGP bacteria" to "PGPB" in line 34.
2. The authors discussed various bacterial genera as biofertilizers in lines 48-53. At the same time, Bacillus, Pseudomonas, and Streptomyces are proven genera with plant growth-promoting bacteria (PGPB) properties. It is advised that the authors address this matter accurately and include appropriate references, such as DOI: 10.3390/agronomy11050846; DOI: 10.3390/mi13091423.
3. English expression and grammar need improvement. Proofread the manuscript before submission. Consult a native speaker for the same.
4. The authors investigated the efficacy of native Sinorhizobium strains in promoting plant growth on common bean plants. It is recommended to use related treatment figures.
5. This research work proposes an intriguing strategy with practical applications. However, it is important for the authors to address the practical considerations for farmers. They should respond to the following questions:
a: How do farmers provide sterilization conditions?
b: Is it necessary for the farmers to be trained to use this technique? Describe how easy or difficult it is to use
c: Is the cost of providing these homemade bioreactors justifiable from the farmer's perspective?
English expression and grammar need improvement. Proofread the manuscript before submission. Consult a native speaker for the same.
Author Response
RESPONSE TO REVIEWER 1
Comments and Suggestions for Authors
The research manuscript titled" Cost-Effective Cultivation of Native PGPB Sinorhizobium Strains in a Homemade Bioreactor for Enhanced Plant Growth" was targeted at the agricultural field. There are some ambiguous that should be considered. After revision, this manuscript can be accepted.
Please answer the following comments in detail:
We would like to express our sincere gratitude to the reviewer for their valuable observations and comments. We have thoroughly reviewed the manuscript and addressed all the points raised by the reviewer. The responses to the observations are indicated in bold black letters. As a result, the document now meets the required scientific and technical standards for publication.
- All abbreviations should be defined at first appearance in the Abstract. Two terms are not defined in the Abstract, such as PY-Ca2+and PGPB. Following this revision, kindly modify "PGP bacteria" to "PGPB" in line 34.
Response 1: In the Abstract section, we have included the definitions of the abbreviations as suggested by the reviewer. For instance, PY-Ca2+ stands for Casein Peptone and Yeast Extract, supplemented with calcium, and PGPB refers to Plant Growth Promoting Bacteria. Similarly, in line 34, we have replaced "PGP bacteria" with "PGPB" as per the reviewer's request. We have ensured accuracy and clarity in complying with the reviewer's recommendations.
- The authors discussed various bacterial genera as biofertilizers in lines 48-53. At the same time, Bacillus, Pseudomonas, and Streptomyces are proven genera with plant growth-promoting bacteria (PGPB) properties. It is advised that the authors address this matter accurately and include appropriate references, such as DOI: 10.3390/agronomy11050846; DOI: 10.3390/mi13091423.
Response 2: We appreciate the reviewer's observation. We have added the appropriate references that support the information. In the References section, we have included the relevant bibliographies along with their corresponding DOIs (Digital Object Identifier).
In the References section, we have added the following references:
[4] Flores‐Félix, José D., Esther Menéndez, Lina P. Rivera, Marta Marcos‐García, Pilar Martínez‐Hidalgo, Pedro F. Mateos, Eustoquio Martínez‐Molina, Ma de la Encarnación Velázquez, Paula García‐Fraile, y Raúl Rivas. 2013. «Use of Rhizobium Leguminosarum as a Potential Biofertilizer for Lactuca Sativa and Daucus Carota Crops». Journal of Plant Nutrition and Soil Science 176 (6): 876-82. https://doi.org/10.1002/jpln.201300116.
[5] Gen-Jiménez A, Flores-Félix JD, Rincón-Molina CI, Manzano-Gomez LA, Rogel MA, Ruíz-Valdiviezo VM, Rincón-Molina FA and Rincón-Rosales R (2023) Enhance of tomato production and induction of changes on the organic profile mediated by Rhizobium biofortification. Front. Microbiol. 14:1235930. doi: 10.3389/fmicb.2023.1235930.
- English expression and grammar need improvement. Proofread the manuscript before submission. Consult a native speaker for the same.
Response 3. Thank you for the observation. The manuscript's English quality has been thoroughly reviewed and edited by MDPI Language Editing Services (English editing ID: English-69741).
- The authors investigated the efficacy of native Sinorhizobium strains in promoting plant growth on common bean plants. It is recommended to use related treatment figures.
Response 4. We have added Supplementary Figure S1, which pertains to the efficacy of native Sinorhizobium strains in promoting plant growth on common bean plants.
- This research work proposes an intriguing strategy with practical applications. However, it is important for the authors to address the practical considerations for farmers. They should respond to the following questions:
a: How do farmers provide sterilization conditions?
b: Is it necessary for the farmers to be trained to use this technique? Describe how easy or difficult it is to use
c: Is the cost of providing these homemade bioreactors justifiable from the farmer's perspective?
Response 5. We appreciate the comments. In the Discussion section, we have included additional information that addresses the questions raised by the reviewer. We believe that we have adequately addressed each of these three questions. “Farmers can use autoclaves and appropriate equipment to sterilize the bioreactor. Providing training on equipment handling and maintenance is crucial. Although homemade bioreactors are user-friendly, farmers may benefit from guidance in growth media preparation and bacterial culture management. Offering technical advice is recommend-ed. The cost of home bioreactors is justified considering the long-term benefits of producing cost-effective biofertilizers, reducing reliance on expensive chemical fertilizers, and promoting sustainable farming. Home bioreactors represent an economically viable and environmentally friendly approach for farmers, especially in vulnerable agricultural sectors”.
Comments on the Quality of English Language
English expression and grammar need improvement. Proofread the manuscript before submission. Consult a native speaker for the same.
Response: Thank you for the observation. The manuscript's structure, editing, and English grammar have been thoroughly reviewed and improved by MDPI Language Editing Services to ensure the quality of this manuscript. English editing invoice:english-69741.

Reviewer 2 Report
There are limited comments about this research paper. Only two points could be more evidenced, in particular for a reader not exactly inside the sector, like, for instance, someone interested in convert the research in a real production. Explain better why the homemade bioreactor should be preferred to those already in use at the industrial level. Report in the conclusion a summary of the cost benefit in using this species in relation to the ordinary media already in use and why it should preferred. Some little corrections are suggested. Pag. 2 line 79 change variable such in variable, such; line 161 change 30°C in 30 °C and the same in pag. 5 line 18 and line 201; pag. 11 line 370 the name of the species should be in Italic.
Author Response
REPONSE TO REVIEWER 2
Comments and Suggestions for Authors
We would like to express our sincere gratitude to the reviewer for their valuable observations and comments. We have thoroughly reviewed the manuscript and addressed all the points raised by the reviewer. The responses to the observations are indicated in bold black letters. As a result, the document now meets the required scientific and technical standards for publication.
There are limited comments about this research paper. Only two points could be more evidenced, in particular for a reader not exactly inside the sector, like, for instance, someone interested in convert the research in a real production. Explain better why the homemade bioreactor should be preferred to those already in use at the industrial level. Report in the conclusion a summary of the cost benefit in using this species in relation to the ordinary media already in use and why it should preferred.
Response: Farmers can use autoclaves and appropriate equipment for sterilizing the bioreactor. Providing training on equipment handling and maintenance is crucial. Although homemade bioreactors are user friendly, farmers may benefit from guidance in growth media preparation and bacterial culture management. Offering technical advice is recommended. The cost of home bioreactors is justified considering long-term benefits of producing cost-effective biofertilizers, reducing reliance on expensive chemical fertilizers, and promoting sustainable farming. Home bioreactors represent an economically viable and environmentally friendly approach for farmers, especially in vulnerable agricultural sectors.
Some little corrections are suggested. Pag. 2 line 79 change variable such in variable, such; line 161 change 30°C in 30 °C and the same in pag. 5 line 198 and line 201; pag. 11 line 370 the name of the species should be in Italic.
Response: We appreciate the reviewer's comments. In the main manuscript, we have made all the corrections suggested by the reviewer. In line 79, the word "variable, such" was corrected. In line 161, "30°C" was changed to "30 °C," and the same correction was made in lines 198 and 201. In line 370, the name of the bacterial species was written in italic as requested.
